# Phenylalanine Residues in the Active Site of CYP2E1 Participate in Determining the Binding Orientation and Metabolism-Dependent Genotoxicity of Aromatic Compounds

**DOI:** 10.3390/toxics11060495

**Published:** 2023-05-31

**Authors:** Keqi Hu, Hongwei Tu, Jiayi Xie, Zongying Yang, Zihuan Li, Yijing Chen, Yungang Liu

**Affiliations:** 1Department of Science and Education, Guangdong Second Provincial General Hospital, 466 Xingang Middle Road, Guangzhou 510317, China; 2Guangdong Provincial Key Laboratory of Tropical Disease Research, Department of Toxicology, School of Public Health, Southern Medical University, Guangzhou 510515, China; 3Guangdong Provincial Center for Disease Control and Prevention, Qunxian Road, Panyu District, Guangzhou 511430, China

**Keywords:** aromatic compounds, CYP2E1, phenylalanine, molecular simulation, random forest

## Abstract

The composition of amino acids forming the active site of a CYP enzyme is impactful in its substrate selectivity. For CYP2E1, the role of PHE residues in the formation of effective binding orientations for its aromatic substrates remains unclear. In this study, molecular docking and molecular dynamics analysis were performed to reflect the interactions between PHEs in the active site of human CYP2E1 and various aromatic compounds known as its substrates. The results indicated that the orientation of 1-methylpyrene (1-MP) in the active site was highly determined by the presence of PHEs, PHE478 contributing to the binding free energy most significantly. Moreover, by building a random forest model the relationship between each of 19 molecular descriptors of polychlorinated biphenyl (PCB) compounds (from molecular docking, quantum mechanics, and physicochemical properties) and their human CYP2E1-dependent mutagenicityas established mostly in our lab, was investigated. The presence of PHEs did not appear to significantly modify the electronic or structural feature of each bound ligand (PCB), instead, the flexibility of the conformation of PHEs contributed substantially to the effective binding energy and orientation. It is supposed that PHE residues adjust their own conformation to permit a suitablly shaped cavity for holding the ligand and forming its orientation as favorable for a biochemical reaction. This study has provided some insights into the role of PHEs in guiding the interactive adaptation of the active site of human CYP2E1 for the binding and metabolism of aromatic substrates.

## 1. Introduction

Biotransformation enzyme-dependent metabolism is one of the most important processes experienced by xenobiotics present in the human body. Among various families of biotransformation enzymes, cytochrome P450s (CYPs) as a superfamily of monooxygenases play a major role in the metabolism of various exogenous and endogenous compounds [1]. The active site of each CYP enzyme is formed by the residues of involving amino acids (AA) and a Fe ion-containing heme group, which provides a hydrophobic cavity and determines the conformation (orientation) of substrate binding [2,3,4]. Studies have shown that increased flexibility of the conformation of the AAs in the active site of CYP102A1 is favorable for reaching an orientation of substrate binding, particularly for a short distance from ligand to the Fe iron in the heme as valid for electron transfer [5]. In human CYP3A4, several AAs in the active site (including ARG 106, ARG 372, GLU 374, PHE 108, and PHE 213) are involved in directing the effective binding of most substrates [6]. Additionally, threonine (THR) 303 in the active site of human CYP2E1 is significant for stable binding of 4-methylpyrazole via forming hydrogen bonds [4]. Subsequent to directed mutations in the active site of human CYP2B4 (F296A, T302A, I363A, and V367L involved), the affinities of various substrates to the enzyme, the rates of product formation, and the potencies of known inhibitors for the mutated CYP2B4 were all tremendously changed, which evidences the importance of those AAs in the catalyzing activity and substrate specificity of the enzyme [7].

The aromatic AAs (e.g., PHEs) in the active site of CYP2E1 have been proven to be of great importance for stabilizing a hydrophobic cavity, sustaining the substrate binding, guiding its orientation, and determining the site of metabolism (SOM) [3,4]. In our previous studies, some phenyl group-containing chemicals appeared to be activated by human CYP2E1 for cytogenetic toxicity [8,9,10,11,12,13]. For example, 1-methylpyrene (1-MP), which is a rodent carcinogen abundantly present in cigarette smoke [14], vehicle emissions [15], smoked or barbecued foods [16], and spills of crude oil [17], has been identified as a substrate for human CYP2E1, with its benzylic hydroxide as the major metabolite [18]. However, CYP2E1 is known to oxidize only small-sized chemicals, such as ethanol and trichloroethylene, since its active site is rather small [4]. As 1-MP has four conjugated benzyl groups, seemingly too large to be hold in the active site of CYP2E1, we were interested in the use of this compound in the docking study for understanding how 1-MP could bind into the small active site as effective for a biochemical reaction, especially for a role of the conformational flexibility of PHEs in the active site.

The molecular size of documented substrates for CYP2E1 seems to vary largely. Benzene and its hydroxylated metabolites (phenol and hydroquinone) are simplest aromatic substrates for human CYP2E1, thereby they are activated for mutagenic effects [8]. However, the PHEs in the active site appear to have little effect on these small ligands during the metabolism process [19]. It is likely that the PHEs in the active site are primarily important for directing the binding of larger (such as multiple phenyl groups-containing) ligands. Polychlorinated biphenyls (PCBs, contain two phenyl rings connected by a single bond), which are a large group of persistent organic pollutants with various health effects, including neuro/behavioral abnormalities [20], endocrine disruptions [21], overweight [22], and cancers [23], are substrates for different CYP2 enzymes giving rise to differently structured products. For example, PCB 95 and PCB 136 may be metabolized by human CYP2B6 to their meta-hydroxylated metabolites, while para-hydroxylated metabolites of PCBs were the major products formed by human CYP2E1 [24]. Interestingly, the amino acid sequences of the active sites of both enzymes are quite similar to each other [3], except that PHE 478 in CYP2E1 is different from the ortholog site in CYP2B6 (VAL477), and it is likely that this discrepancy is involved in forming varying orientations of PCBs in the different active sites.

Collectively, the importance of PHEs in the active site of a CYP enzyme for substrates containing varying number of phenyl groups may vary in some way. To gain deeper insights into the role of PHEs in directing the binding and effective orientation of aromatic substrates in the active site of human CYP2E1, computational simulation methods such as molecular docking, molecular simulation, quantum chemistry approaches, and random forest model were employed in this study. Meanwhile, based on the established documents of human CYP2E1-activated mutagenicity of aromatic compounds in mammalian cells, a preliminary exploration of the screening value of the ligand-enzyme molecular docking analysis in the prediction of human CYP2E1-activated promutagens as candidate compounds to be tested in subsequent genotoxicity assays, was also conducted.

## 2. Materials and Methods

### 2.1. Preparation of Protein and Ligand Structures

The primary 3D structures of human CYP2E1 were obtained from RCSB Protein Data Bank (3LC4, 3KOH, 3GPH, 3E4E, 3T3Z, and 3E6I for CYP2E1). Ligands of this 3D model were removed before protein energy minimization. Charmm36 force field was used for all protein residues [25], and the TIP3P model was applied for water molecules [26]. Energy minimization was performed by Gromacs (Version 2018.4) [27] using the steep algorithm with 0.01 nm for step size and was stopped when the maximal force was less than 1000.0 kJ/mol/nm. The 3D structures of ligands were built by Avogadro [28] and preliminarily optimized by using GFN2-XTB method [29].

### 2.2. Molecular Docking

Molecular docking was performed by using Autodock version 4.2 suits [30]. Before docking, water molecules were removed from the protein structures. Gasteiger charges were added to the protein and ligands. The grid box was set to cover the active zone of proteins. The center of grid was on element Fe in the heme and the grid point values of *x*-, *y*-, and *z*-axes were set to 60, 60, and 60 with a spacing of 0.375 Å (1 Å is equal to 0.1 nm). Lamarckian genetic algorithm (LGA) [31] was performed for ligand-protein docking calculations. The default parameters of LGA were used and the number of LGA runs was set to 50. The 50 conformations were clustered by using a root-mean-square deviations (RMSD)-tolerance of 2.0 Å, then the typical conformation of each cluster was analyzed. The best conformation of protein-ligand complex was selected on the basis of mean binding energy score, the number in each cluster, and the distance between a ligand and the heme.

### 2.3. Molecular Equilibration, Dynamic and Free Energy Calculation

The best conformation of each ligand-protein complex as obtained from molecular docking was further used for molecular dynamics analysis. Protein and ligands were described by Charmm36 force field and CgenFF (the official CHARMM general force field server), respectively [25,32]. The canonical molecular dynamics with periodic boundary condition was performed by using GROMACS (Version 2018.4), with parameters being set according to previous descriptions [19]. Briefly, short-range nonbonded interactions were cut off at 1.2 nm, with long-range electrostatics calculated using the particle mesh Ewald (PME) algorithm. The bonds to H were constrained with LINCS algorithm. After energy minimization, isothermic-isopyknic ensemble (NVT) for 100 ps was performed, of which temperature was maintained at 310 K. Following NVT equilibration, isothermic–isobaric ensemble (NPT) for 100 ps was performed and the pressure was maintained at 1.0 bar. For both NVT and NPT equilibration, the position of ligands was restrained, and modified Berendsen thermostats were set for protein-ligand complex and other components. Finally, the production MD simulations were conducted for 50 ns without any restraints. Modified Berendsen thermostat and Parrinello-Rahman were applied to maintain temperature and pressure, respectively. The time step of integration was 2 fs and the energy and coordinates of the system were recorded every 10 ps.

Molecular Mechanics/Poisson-Boltzmann Surface Area (MMPBSA) method was used to estimate the free energies of protein-ligand complex [33]. In this study, MMPBSA calculation was performed by using gmx_mmpbsa (version 2022.11.19) [34]. Five hundred snapshots of each complex were extracted from the last 10 ns of molecular simulations.

### 2.4. Tunnel Analysis

As ligands need to pass tunnels to reach the active site of any enzyme, Caver 3.0 software was applied to identify the potential tunnels leading to the active site of human CYP2E1 [35]. Five hundred snapshots were sampled from the molecular trajectories of 1-MP-CYP2E1 complex during the last 10 ns. The heme group was set as the starting point for tunnel searching. The probe radius and the clustering threshold was 1.0 and 4.5 Å, respectively. The other parameters were established using default settings. After calculations, tunnels were visualized by VMD (version 1.9) [36] and the parameters of the tunnel bottleneck were collected for further investigation.

### 2.5. GFN2-xTB- and DFT-Based Simulations

For a semi-empirical description of the each PCB binding to the active site of CYP2E1 (or its F478V mutant), the GFN2-xTB method was applied to perform geometry optimization and calculate the fractional occupation density analysis (FOD) [37]. The analytical linearized Poisson-Boltzmann (ALPB) model was employed to provide the implicit solvation in chloroform for the total system.

The quantum mechanics calculation was performed by using ORCA 5.0 [38]. The structure of each PCBs and its PHE complexe were optimized by using the unrestricted hybrid B3LYP functional [39] in combining with the def2-svp basis set [40]. The wave functions and more accurate energies of the N, N + 1, and N − 1 electronic states were obtained using single-point calculations with PWPB95 functional [41] and the def2-TZVPP basis was set for the optimized structures. Since the B3LYP functional cannot well describe the medium to long range dispersion interactions of the reaction system, D3-BJ correction developed by Grimme was used to correct the dispersion calculation in all systems [42,43]. In addition, structure optimization and frequency calculations were performed synchronously to confirm that the structure was in the ground state.

### 2.6. Random Forest Model

The dataset used in this study includes 33 PCBs with 19 features (Appendix A) to build the binary classification model for predicting the human CYP2E1-dependent mutagenicity of PCBs. These features were collected from molecular docking, GFN2-xtb calculation, DFT calculation and CompTox Chemicals Dashboard [44]. 30% of the data were randomly sampled from the dataset and set as a test dataset to quantify the prediction performance of the developed Random Forest models [45]. The number of trees in the ensemble (*ntree*) was set to 1000 and the terminal node size (*nodesize*) was 5. The number of randomly selected predictor variables at each node (*mtry*) was determined by grid search tuning based on lowest RMSE (root-mean-square error) with a 5-fold cross-validation and 20 repeats [46,47]. To determine the performance of the model the mean values of the parameters (including accuracy, precision, recall rate, and F1 score) were determined.

## 3. Results

### 3.1. The Significance of Conformational Flexibility of PHE Residues for the Affinity of Aromatic Substrates to Human CYP2E1

As previously reported, under a molecular docking condition with the residues of AAs forming the active site of human CYP2E1 set as rigid conformations, the results of the docking of benzene and its oxidized products to the enzyme are in good agreement with the experimentally obtained positive genotoxicity results in mammalian cell lines genetically engineered for stable expression of human CYP2E1 [19]. It appears that setting PHEs as flexible residues is generally unnecessary for docking small-sized ligands to human CYP2E1. However, bulky ligands like 1-MP, which have four conjugated phenyl rings, cannot be docked into the active site of human CYP2E1 under rigid docking settings, as shown in the (Appendix A, based on the structure of 3E6I). Therefore, PHE 478, which plays an important role in forming some tunnels in the human CYP2E1 for substrate binding, was set as a flexible residue so that a larger cavity as the active site can be formed. As a result, 1-MP was successfully docked into the active site of human CYP2E1, with a binding energy of −9.99 kcal/mol, where the conformational flexibility of PHE 478 residue (through rotating and moving) provided sufficient space for the entry of 1-MP (Figure 1B). However, the orientations of the benzyl group in 1-MP, which has been proved to be the main hydroxylation site, did not permit a proper distance (<6.0 Å) to heme (actually 10.4 Å).

To further understand the contribution of PHEs to effective binding of 1-MP to human CYP2E1, all the PHEs (including PHE 106, PHE 115, PHE 207, PHE 298, and PHE 478) in the active site of human CYP2E1 were set as flexible residues. As shown in Figure 1C, the binding of 1-MP was thus in a favorable orientation for its hydroxylation at the benzyl group. The PHE residues induced a way of 1-MP binding to the active site by rotation and forming π–π bonds to stabilize the ligand binding. Moreover, similar docking results optimized by setting flexible PHEs were observed in other CYP2E1 structures derived from different PDB sources (Appendix A).

### 3.2. Impact of PHE 298 and PHE 478 Residues on the Binding of 1-MP to Human CYP2E1 and Its Orientation

Tunnel analysis showed that tunnels 2b,c were favorable for 1-MP to approach the active site of human CYP2E1 (Appendix A, Appendix A). As PHE 298 and PHE 478 are important controllers of the “gate” of 2c and 2b tunnel, respectively [4,48], we constructed mutated protein models with alanine residue at both sites (F289A and F478A), then the role of these PHEs in the binding of 1-MP to the enzyme was analyzed. The docking results indicated that 1-MP was able to bind to the active site of each mutant CYP2E1, but with a lower affinity than that of the wild enzyme (Table 1). Further molecular dynamics simulations were performed based on the best conformations obtained from molecular docking. As shown in Figure 2A, the RMSD of each complex was gradually convergent and the system remained in equilibrium status during the last 40 ns. However, the distances between 1-MP (Cα atom) and the heme (Fe ion) differed greatly in various complexes. Illustrated in Figure 2B, 1-MP stably bound to the wild-type model with distances around 5 Å, while its binding to F298 mutant was unstable, with long distances (mostly >10 Å) from the heme; and in case of F478A mutant drastic fluctuation in the distance was observed, making stable binding impossible.

The last 10 ns of each complex’s trajectory were extracted for further MMPBSA calculations. As shown in Table 1, 1-MP was tightly bound to the active site of each CYP2E1 model, but the binding energy of each 1-MP-mutant protein complex was less negative than that of 1-MP-wild protein complex. Moreover, the energy contributions from the residues at the active site of the mutant CYP2E1 models differed from that of the wild type model (Figure 3 and Appendix A). Although the energy contributions from PHE 106 and PHE 116 were slightly increased (with that from PHE 207 and PHE 298 unchanged) in both mutants, mutations on PHE 478 caused a steep energy change, which suggests an important role of PHE 478 in the ligand binding and orientation (Figure 3).

### 3.3. Consistency between Molecular Simulation Results and Historical Data of Human CYP2E1-Activated Mutagenicity of PCBs in Genetically Engineered Mammalian Cells

Based on previous experimental studies, a series of PCBs are metabolically activated by human CYP2E1 for mutagenic effects in mammalian cells. Therefore, this study utilized a random forest model that incorporated various factors, such as flexible molecular docking of PHEs, semi-empirical calculations of PCB−CYP2E1 complex, conceptual density functional parameters of PCB−PHE complex, and the physical/chemical property of each PCB (such as molecular weight, chlorine substitution pattern, octanol-water partition coefficient, concentrations of hydroxylated PCB metabolites in air, and their half-lives in fish) to investigate the association between these factors and human CYP2E1-activated PCB mutagenicity, as well as the role of PHEs in the activating process (Appendix A).

The results indicated that the random forest model, which was constructed by taking multiple factors into consideration, was able to predict human CYP2E1-dependent PCB mutagenicity with some accuracy, as demonstrated by a coincidence rate of 79%, a precision of 79.3%, a recall of 90%, and an f1 score of 0.838. Furthermore, a detailed analysis of the importance of each factor indicated that some factors, such as the ligand-to-heme distance and docking score obtained by molecular docking, the nucleophilicity index (based on conceptual density functional theory), HOMO−LUMO gap of the ligand, and half-life in fish, were significant determinants for predicting the human CYP2E1-dependent PCB mutagenicity (Figure 4). Among these factors, the flexible molecular docking of PHEs exhibited a dominant influence. Conversely, through quantum chemistry and semi-empirical analysis PHEs were not significantly influential on the intrinsic properties of PCBs, which suggests that PHEs may play an important role in regulating the binding of each PCB to the active site rather than affecting the electronic/chemical properties of ligands.

## 4. Discussion

The steric shape of the cavity and the AA composition of the active site are important factors determining the substrate selectivity of an enzyme. In the active sites of the CYP1 enzymes a PHE cluster exists above the heme, thereby they tend to metabolize planar substrates such as polycyclic aromatic hydrocarbons, dioxins, and dioxin-like PCBs [49,50]. The substrate selectivity of CYP2 isoforms varies greatly, among which CYP2A and 2B prefer neutral compounds as their substrates, where the substrates of CYP2B are generally more lipophilic than those of CYP2A. CYP2C and CYP2D, which contain relatively low numbers of PHEs in their active sites, are more likely to catalyze the metabolism of weakly acidic and weakly basic substances, respectively [49,51]. The differences in the number and position of PHE residues may affect the selectivity for, and the affinity of, substrates. Studies have shown that in human CYP3A4, PHE108, PHE213, PHE220, and PHE241 are involved in the allosteric effects in F helix/G helix region, resulting in an extension of helix F and a larger active site; while in CYP3A5, the replacement of PHE at position 108 with LEU leads to changed plasticity of the F/G helix region, producing a substrate-binding pattern different from that in CYP3A4 [52]. After mutating the HIS85 of CYP101B1 to PHE, there appeared several changes: significantly increased rate of the binding of hydrophobic substrates to the enzyme, decreased affinity for β-ionone, and shifted site of oxidation [53]. For CYP2E1, the high proportion of PHE residues in its active site may also play an important role in its selectivity of substrates and their binding orientation [4].

For benzene and its oxidized metabolites, which are known mutagens activated by human CYP2E1, the docking results were highly consistent with their mutagenic effects observed in vitro [19]. Benzene exhibited the strongest affinity toward the enzyme, while the ultimate mutagen 1,4-benzoquinone was difficult to enter the active site of CYP2E1. As indicated in this study, since benzene and its hydroxylated products are relatively small compared to aromatic compounds containing two and more phenyl groups, the PHEs in the active site of CYP2E1 did not show significant regulation on the orientation of the small ligands. On the other hand, under rigid docking conditions, the space of the active site of human CYP2E1 is too small to accommodate larger molecules which contain multiple benzene rings (such as 1-MP), but this barrier (caused by limited cavity size) can be eliminated by increased conformational flexibility of PHEs [4,54]. In vitro tests indicated that human CYP2E1 can bind to substrates via ligand recognition and induced fit [55]. Furthermore, molecular dynamics simulations showed that human CYP proteins can make conformational change through the activities of the F/G loop, B/C loop, and corresponding α helices, thus the volume of the active site of CYP2E1 can expand from 220 Å^3^ to 1310 Å^3^ (when F/G loop is fully open) [55].

Some AAs play an important role in the conformational changes and cavity expansion in CYP enzymes, among which PHE 478 or PHE at other positions (such as PHE 476 in human CYP2C9) are important “gatekeeper” amino acids for regulating the entry of substrate to, and its exit from, the active site of each CYP enzyme, and they have an important impact on the long-range molecular interactions between substrates and CYPs [4,56]. In this study, via molecular simulation two PHE mutants of CYP2E1 were built for the analysis of the potential functions of PHE298 and PHE478 in substrate binding and orientation. The results showed that the mutation of either PHE residue had a negative impact on the binding affinity and orientation (for catalyzing a biotransformation reaction) of 1-MP to CYP2E1, and the mutation of PHE478 showed the most significant impact on the binding status. In addition, it is predicted that the rotation of PHE 478 can connect the catalytic site to the secondary binding region and form a larger space to hold bulky substrates [54]. Therefore, the conformational changes of PHE 478 in human CYP2E1 might be necessary for the access of large-sized substrates, including 1-MP and PCBs, to the active site. Our docking results showed that after the rotation of PHE 478 to some extent, the cavity of the active site can be enlarged, allowing these bulky molecules to bind to the active site of human CYP2E1. Nevertheless, even under flexible settings for PHE478 the benzylic group of 1-MP is still too far from the heme to permit a biochemical reaction [13]. After setting all PHE residues in the active site as flexible, most conformations showed that the benzylic group became closer to the heme, which is consist with the observed production of benzylic hydroxylation by the enzyme [18]. Therefore, our results suggest that the flexibility of PHEs may be critical for catalyzing reactions of aromatic substrates, as the aromatic AA residues in the active site can easily form non-covalent interactions such as π–π interaction with ligands, thus provide a sufficiently large cavity for the binding of large substrates, and further adjust their orientation [54,57,58].

It is established that the interactions between the residues of some AAs and a substrate may affect the process of the transfer of proton/electron, thereby enhancing their reactivity of the substrate and/or reducing the barrier from a reaction [59,60]. This study has further explored potential factors involved in substrate-enzyme interaction for the catalysis of biotransformation reactions, by using a model constructed with data calculated in various ways. The results indicated that some factors with the substrates, including nucleophilicity, HOMO−LUMO gap, in vivo half-life, and molecular docking results based on flexible setting of PHE conformation, contributed substantially to the prediction of human CYP2E1-activated mutagenicity of PCBs. It looks that PHEs in the active site of human CYP2E1 may participate in locating the site of reaction and the formation of products by enlarging cavity size for holding bulky substrates and regulating their orientations. However, the list of PCBs in this study was not based on random choice, but was guided by ever observed structure-genotoxicity relationships [10,11,12,13], thus a bias may have been introduced, leading to relatively high predictive value of molecular docking for human CYP2E1-activated PCB mutagenicity [12]. Therefore, in the future it is necessary to use substrates of random and more diverse structural patterns in the study of PHE−CYP2E1 interaction, to further identify the role of PHEs in regulating the metabolism of aromatic substrates by human CYP2E1 and the prediction of metabolically activated toxicity.

## Figures and Tables

**Figure 1 toxics-11-00495-f001:**
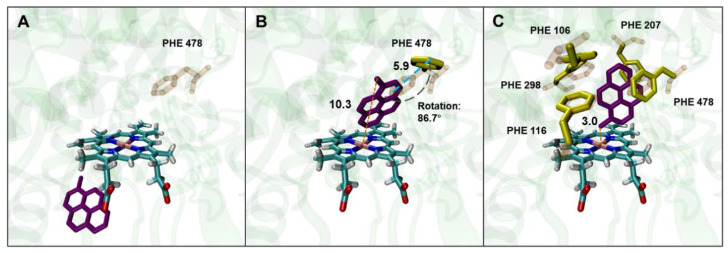
Conformations of 1-MP docked into human CYP2E1 under different settings for PHE residues. (**A**): Docking with rigid settings for all AA residues in the active site; (**B**): PHE 478 being set as a flexible residue in the active site for docking; (**C**): all PHEs in the active site being set as flexible residues for docking. Orange dotted lines represent distances between the Cα atom of 1-MP and Fe ion (Å); the blue dotted line represents the range of the movement of PHE 478 (Å); the black dotted line represents the rotation of PHE 478.

**Figure 2 toxics-11-00495-f002:**
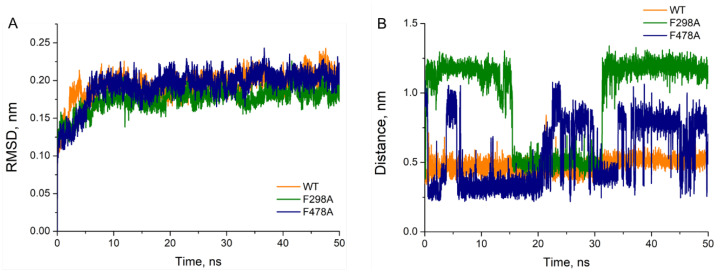
RMSD (**A**) of wild-type and mutant human CYP2E1 protein bound with 1-MP and the variation of distance between the Cα in 1-MP and the heme in a period of 50 ns (**B**).

**Figure 3 toxics-11-00495-f003:**
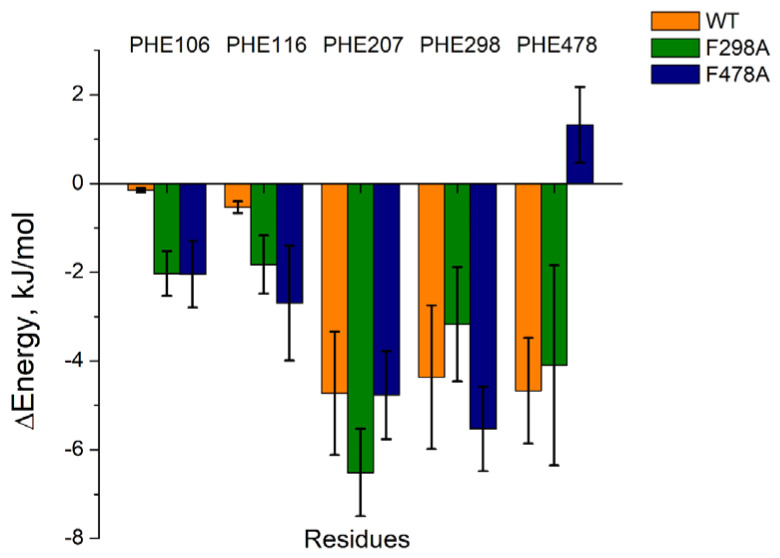
The energy contributions from different PHEs in the active site of 1-MP−wild−type CYP2E1 and that of each 1-MP−mutant CYP2E1 complex.

**Figure 4 toxics-11-00495-f004:**
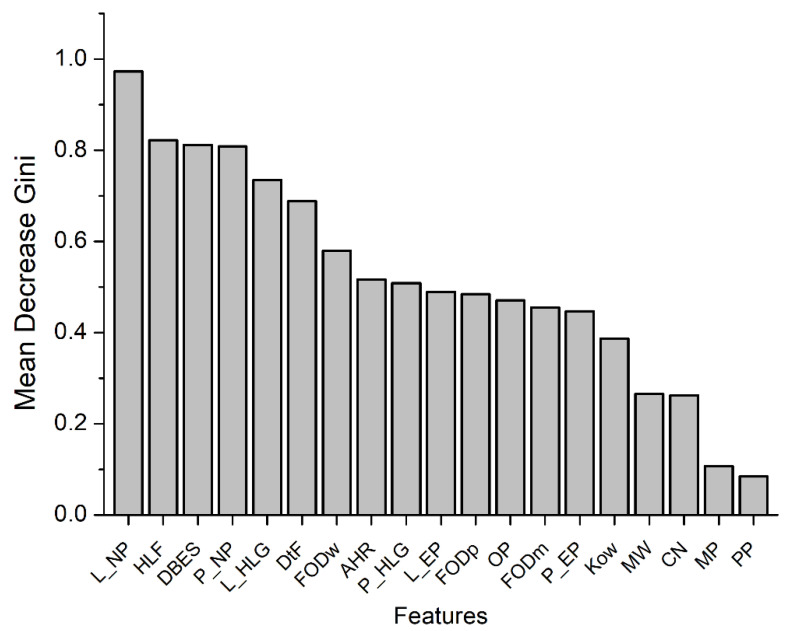
Relative importance of each molecular feature of PCBs in the random forest model. L_NP, the nucleophilicity of ligand; HLF, fish biotransformation half-life; DBES, binding energy score obtained from molecular docking with PHE478 set flexible; P_NP, the nucleophilicity of ligand-PHE complex; L_HLG, the HOMO-LUMO gap of ligand; DtF, the distance from SOM to Fe ion in heme; FODw, the FOD value of ligand binding to the active site of human CYP2E1; AHR, atmospheric hydroxylation rate; P_HLG, the HOMO-LUMO gap of ligand-PHE complex; L_EP, the electrophilicity of ligand; FODp, the FOD value of ligand; OP, the number of ortho-Cl-substitution; FODm, the FOD value of ligand binding to the active site of F478A mutant; P_EP, the electrophilicity of ligand-PHE complex; Kow, LogKow: octanol-water partition coefficient; MW, the molecular weight; CN, the number of Cl-substitution; MP, the number of meta-Cl-substitution; PP, the number of para-Cl-substitution.

**Table 1 toxics-11-00495-t001:** Free energy of 1-MP for binding into the active site of wild type and mutant human CYP2E1 protein.

CYP2E1 Models	Energy for 1-MP Binding to the Active Site of Wild-Type/Mutant Human CYP2E1, kJ/mol				
	Coulombic	Lennard-Jones	Polar Solvation	Non-Polar Solvation	Total
WT	−7.55 ± 1.08	−148.50 ± 7.43	31.59 ± 2.63	−16.75 ± 0.32	−141.20 ± 7.65
F298A	−1.92 ± 1.77	−135.69 ± 7.86	34.12 ± 2.28	−17.20 ± 0.20	−120.68 ± 7.56
F478A	−2.12 ± 1.83	−134.83 ± 7.35	44.45 ± 5.21	−17.52 ± 0.37	−110.02 ± 9.35

## Data Availability

Research data are available upon request from readers.

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
