# Peer review of "Phenylalanine Residues in the Active Site of CYP2E1 Participate in Determining the Binding Orientation and Metabolism-Dependent Genotoxicity of Aromatic Compounds"

_toxics, 2023, doi:10.3390/toxics11060495_

Round 1

Reviewer 1 Report

The manuscript  Phenylalanine Residues in the Active Site of CYP2E1 Partici-2 pate in Determining the Binding Orientation and Metabolism-3 dependent Genotoxicity of Aromatic Compounds.” by Keqi Hu et al. reports a molecular docking and molecular dynamics analysis toward the interactions between Phenylalanines in the active site of uman CYP2E1 and various aromatic compounds as confirmed CYP2E1 substrates. Results have provided some insights into the role of Phenylalanines in managing the interactive adaptation of the active site of human CYP2E1 for the binding and metabolism of aromatic substrates. The authors have been involved in studying the CYP enzyme for a long time.

This study can be published although results do not have relevant impact.

Author Response

Reviewer #1

Question 1: This study can be published although results do not have relevant impact.

Answer: We admit that the methodology of molecular docking and molecular dynamics analysis may currently beyond the understanding capacity of most audience of this journal, however, digestion of the toxicological significance of predicting the substrate potentials of chemicals for relevant biotransformation enzymes is relatively easy and interesting; moreover, from a developmental perspective some readers of our article may become motivated to integrate molecular simulation methods into their future research work. It looks that our work will have some impact in the era with predictive toxicology becoming more and more popular.

Reviewer 2 Report

GENERAL COMMENTS

Hu and colleagues have presented an in silico approach aimed at assessing the role of PHEs in the active site of CYP2E1 and how it affects to the metabolism-dependent genotoxicity of aromatic compounds. The authors performed a molecular docking and molecular dynamic analysis. Moreover, a random forest model was used to study the genotoxicity of polychlorinated biphenyls. Results indicated that flexibility of PHEs plays an important role to permit the binding energy and orientation of these compounds.  

The methods of analysis appear appropriate, and the results do not appear to be over-interpreted. However, I only have few general comments, and I would suggest a minor revision of the manuscript prior publication.

SPECIFIC COMMENTS FOR REVISION

-I have found some typographical errors in the document. I point out the following, but there are likely others, so a closer read of the manuscript is warranted:

Please check abbreviations along the manuscript:

- Abstract. Line 21: the abbreviation PCB is used and then it is described at line 23.

-Introduction. Line 39: amino acids are abbreviated as AA; however, the abbreviation AA is not used in the text anymore. Please use abbreviation or eliminate AA.

            -Introduction. Line 62: Capital letters after period. “benzene” should be “Benzene”.

-Results. Line 237. Again, polychlorinated biphenyls is described and abbreviated.

- Too much information is given at footnotes. For eg in Figure 1, all results are again explained at the footnotes. I recommend shortening them, since the information is written at the manuscript.

 -Sufficient information is given at the introduction section. However, I recommend to the authors to add more information about the exposition of humans to PCBs (and specially to 1-MP). I found this information relevant since they are evaluating the effect in human CYP2E1.

-The reason why authors chose 1-MP for this study remains unclear. It is stated that CYP2E1 is known to oxidize small xenobiotic compounds however, 1-MP (which is a bulky substrate) has been identified as a substrate for this enzyme. Is that the reason why 1-MP has been selected for this study? Please, clarify at introduction section.
Is there any other compound that could have been used?

-  Author concluded that further studies are necessary testing PCBs with higher degrees of chlorination. The level of chlorination is not a parameter discuss in the manuscript. Could they explain how it can affect with the interaction with PHE-CYP2E1 and its mutagenicity?

Overall english is correct however, some typographical mistakes can be found in the manuscript. 

Author Response

Reviewer #2

GENERAL COMMENTS

Hu and colleagues have presented an in silico approach aimed at assessing the role of PHEs in the active site of CYP2E1 and how it affects to the metabolism-dependent genotoxicity of aromatic compounds. The authors performed a molecular docking and molecular dynamic analysis. Moreover, a random forest model was used to study the genotoxicity of polychlorinated biphenyls. Results indicated that flexibility of PHEs plays an important role to permit the binding energy and orientation of these compounds.  

The methods of analysis appear appropriate, and the results do not appear to be over-interpreted. However, I only have few general comments, and I would suggest a minor revision of the manuscript prior publication.

SPECIFIC COMMENTS FOR REVISION

-I have found some typographical errors in the document. I point out the following, but there are likely others, so a closer read of the manuscript is warranted:

Please check abbreviations along the manuscript:

Question 2: - Abstract. Line 21: the abbreviation PCB is used and then it is described at line 23.

Answer: As suggested, we have defined PCB with its full term at its first appearance in the Abstract in our revised manuscript. Please see line 22.

Question 3: -Introduction. Line 39: amino acids are abbreviated as AA; however, the abbreviation AA is not used in the text anymore. Please use abbreviation or eliminate AA.

Answer: As suggested, we have modified “amino acid” to “AAs”, and used this abbreviation throughout the text of the revised manuscript. Please see line 39 and the rest of the text in our revised manuscript.

Question 4: -Introduction. Line 62: Capital letters after period. “benzene” should be “Benzene”.  

Answer: This has been changed as suggested. Please see line 68 in the revised manuscript.

Question 5: -Results. Line 237. Again, polychlorinated biphenyls is described and abbreviated.

Answer: This has been changed by replacing the full spellings with PCBs. Please see line 248 in the revised manuscript.

Question 6: - Too much information is given at footnotes. For eg in Figure 1, all results are again explained at the footnotes. I recommend shortening them, since the information is written at the manuscript.

Answer: As suggested, we have shortened the legend of Figure 1. Please see lines 206-210.

Question 7: -Sufficient information is given at the introduction section. However, I recommend to the authors to add more information about the exposition of humans to PCBs (and specially to 1-MP). I found this information relevant since they are evaluating the effect in human CYP2E1.

Answer: As suggested, we have added some information about the exposure of PCBs and 1-MP in human environment. Please see line 57-59 and 73-76

Question 8: -The reason why authors chose 1-MP for this study remains unclear. It is stated that CYP2E1 is known to oxidize small xenobiotic compounds however, 1-MP (which is a bulky substrate) has been identified as a substrate for this enzyme. Is that the reason why 1-MP has been selected for this study? Please, clarify at introduction section.

Answer: Indeed, 1-MP was chosen as it is a bulky substrate with four conjugated phenyl groups, which was large enough to test the potential flexibility of PHEs in the active site of human CYP2E1 for ligand binding. We modified the sentence “As 1-MP has four conjugated benzyl groups, seemingly too large to be hold in the active site of CYP2E1, we were interested in the use of this compound in the docking study for understanding how 1-MP could bind into the small active site as effective for a biochemical reaction, especially for a role of the conformational flexibility of PHEs in the active site.”  to make this point clear. Please see Line 62-66.

Question 9: Is there any other compound that could have been used?

Answer: In some recent reports, including those by our group, some organophosphorus flame retardants, which have two or three phenyl groups (with molecular sizes close to 1-MP), appeared to be metabolized and activated by human CYP2E1 for mutagenic effects in mammal cells (doi: 10.1016/j.tox.2022.153175, doi: 10.1016/j.envpol.2021.117527). But they were not used in this study, since 1-MP has the largest size among substrates of human CYP2E1 and with more comprehensive existing data of metabolism and catalyzing CYP enzymes, thus it could be representative of other large-size substrates of the enzyme in the study on the interactions between ligand and PHE residues.

Question 10:-  Author concluded that further studies are necessary testing PCBs with higher degrees of chlorination. The level of chlorination is not a parameter discuss in the manuscript. Could they explain how it can affect with the interaction with PHE-CYP2E1 and its mutagenicity?

Answer: As suggested, the level of chlorination is not an important parameter in this study, so we have removed the indicated sentence from our revised manuscript.

Reviewer 3 Report

The current study is a pretty comprehensive computational evaluation of the role of phenylalanines in the active site of a human detoxification protein CYP2E1.  The methods appear to be well described but I am wondering about the interest to Toxics readership might have on this paper.  In my view it should have been submitted to a computational journal.

Also the paper is badly written, Figures S1 is not cited in the text.  The authors talk about phenylalanine residue (PHE) binding as if they are free ligands, they use several PDB structures of the mono-oxygenase without explanation.  There is no "flexible docking of PHE" since such residue is covalently bound to the protein. The CYP2E1 is not even introduced so it took me reading the entire paper to understand what this protein was supposed to do.  It is a mono-oxygenase that renders the PCB substrates more polar, which allow them to be excreted but at the same time become carcinogenic (no explanation for that). 

Figure 1 is hard to read and appears to show a weak interaction with the site of oxidation.  The caption is not clear (what is Angstroms doing in parenthesis?) and it is included in the text!  Same goes for Figure 4, I wish they would write the descriptors in the order of importance in the caption.  Also since the half life of the compounds is the second factor in determining the Ginni factor (this was not explained, although it makes sense), I don't see how the calculations done here have a predictive value for toxicity studies. 

The MDs shown in Figure 2 are interesting and consistent with the docking experiments.  But these are of limited utility since it is likely the binding of MP-1 occurs on the timescale of milliseconds.

Figure 3 is incomprehensible There are three Phe residues in the headers that make sense (106, 116, and 207) but what is the effect of Phe 298 on the F298A and the same for Phe 478? It does not make sense.

The authors mention throughout the effect of AA residues on the electronic structure of the PCB substrates.  But there is not spectroscopy literature to back up that claim.  Distortion of ground states in substrates has not really been proven as source of catalytic activity, mostly because the energies involved are not reachable by standard protein dynamic motions.   Sure reactive substrates (nucleophiles in this case) will react faster, but the other factors such as HOMO-LUMO energy alterations and other calculations performed here in do not appear to have a major effect on binding and/or complex reactivity

English could be a lot better.  I can follow the meaning of the sentences but only after reading them several times.  The text requires extensive editing.

Author Response

Question 1. The current study is a pretty comprehensive computational evaluation of the role of phenylalanines in the active site of a human detoxification protein CYP2E1. The methods appear to be well described but I am wondering about the interest to Toxics readership might have on this paper. In my view it should have been submitted to a computational journal.

Answer: This question has been answered in our response to Question #1.

(Question 1: This study can be published although results do not have relevant impact.

We admit that the methodology of molecular docking and molecular dynamics analysis may currently beyond the understanding capacity of most audience of this journal, however, digestion of the toxicological significance of predicting the substrate potentials of chemicals for relevant biotransformation enzymes is relatively easy and interesting; moreover, from a developmental perspective some readers of our article may become motivated to integrate molecular simulation methods into their future research work. It looks that our work will have some impact in the era with predictive toxicology becoming more and more popular.)

Question 2. Also the paper is badly written, Figures S1 is not cited in the text.

Answer: As suggested, we cited Figure S1 in the revised manuscript. Please see line 214. In response to the criticism on the quality of English writing in this manuscript, we have checked through the fulltext thoroughly and made intensive editing. Please check through the revised manuscript for its quality of writing.

Question 3. The authors talk about phenylalanine residue (PHE) binding as if they are free ligands, they use several PDB structures of the mono-oxygenase without explanation. There is no "flexible docking of PHE" since such residue is covalently bound to the protein.

Answer: The PHEs were not regarded as free ligands during the molecular simulations. Actually, the PHEs still formed peptide bonds with its neighbor amino acids, only the single bond connecting R-group of PHE to the Cβ atom and that connecting different atoms within the R-group can be rotated to some extent permitted by the interactive neighboring residues of nearby amino acides. All the PDB structures are the structures of CYP2E1 bound with ligands of varying sizes. Since the size of the ligands may also interactively affect the shape/size of the active site of CYP2E1, the results of molecular docking with each PDB structures may also reveal the degree of conformational flexibility of PHEs that determines the fitting of differently sized substrates into the active site of human CYP2E1.

Question 4. The CYP2E1 is not even introduced so it took me reading the entire paper to understand what this protein was supposed to do. It is a mono-oxygenase that renders the PCB substrates more polar, which allow them to be excreted but at the same time become carcinogenic (no explanation for that).

Answer: Although oxidative metabolism of PCBs by CYPs produce metabolites more polar than their proto compounds, their bio-reactivity may be increased, and they may undergo further oxidation process by CYP enzymes. In our previous studies, the CYP2E1-dependent mutagenicity of some PCBs was observed in metabolically competent mammalian cell lines. Thus, the details of the enzyme-dependent chemical mutagenicity need to be investigated. As suggested, we have added brief introduction to CYP2E1, for better understandability of CYP2E1-associated topics. Please see line 53-84 in the revised manuscript.

Question 5. Figure 1 is hard to read and appears to show a weak interaction with the site of oxidation. The caption is not clear (what is Angstroms doing in parenthesis?) and it is included in the text!

Answer: As indicated in our response to Question 6, we have shortened the legend of Figure 1, and revised it for better clarity. Angstrom is a length unit (1 Å is equal to 0.1 nm), which is commonly used in molecular simulation. As suggested, we have added an explanation for this in the text. Please see line 110-111.

Question 6. Same goes for Figure 4, I wish they would write the descriptors in the order of importance in the caption.

Answer: As suggested, we have re-ordered the descriptors based on their importance. Please see line 274-283.

Question 7. Also since the half life of the compounds is the second factor in determining the Ginni factor (this was not explained, although it makes sense), I don't see how the calculations done here have a predictive value for toxicity studies.

Answer: As a lower half-life indicates increased rate of ligand metabolism, thus the metabolism-associated half-life for each PCB (rather than the compound itself) as a descriptor might be predictive of its specific CYP enzyme-dependent mutagenicity.

Question 8. The MDs shown in Figure 2 are interesting and consistent with the docking experiments. But these are of limited utility since it is likely the binding of MP-1 occurs on the timescale of milliseconds.

Answer: The MDs we performed did not mean to present the process of ligands entering the active site of CYP2E1, which indeed occurs on the timescale of milliseconds. However, the timescale of nanoseconds is long enough for simulating the binding status of 1-MP with wild-type or mutant CYP2E1 and for studying the interactions between ligands and PHEs in the active site of the enzyme. Thus, the simulation period for MD is sufficient in this study.

Question 9. Figure 3 is incomprehensible There are three Phe residues in the headers that make sense (106, 116, and 207) but what is the effect of Phe 298 on the F298A and the same for Phe 478? It does not make sense.

Answer: All the PHEs demonstrated negative binding energies in wild-type model, while in the mutant models (with PHE298 or 478 mutated into ALA298 or 478, respectively) the intensity of negative energy produced by PHE106, 116 and 207 (in F298A) were increased, while that by ALA298 (in F298A model) and ALA478 (in F478A model) were reduced, indicating that PHE298 and PHE478 might be critical AA residues for ligand binding in the wild-type model.

Question 10. The authors mention throughout the effect of AA residues on the electronic structure of the PCB substrates. But there is not spectroscopy literature to back up that claim. Distortion of ground states in substrates has not really been proven as source of catalytic activity, mostly because the energies involved are not reachable by standard protein dynamic motions. Sure reactive substrates (nucleophiles in this case) will react faster, but the other factors such as HOMO-LUMO energy alterations and other calculations performed here in do not appear to have a major effect on binding and/or complex reactivity

Answer: Thank the reviewer for the guidance. Although no spectroscopy literature is available to back up that claim, some quantum mechanics studies indicated that the residues around the active site may affect the reactivity of ligands. Thus, we have modified the relevant sentence to “It is established that the interactions between the residues of some AAs and a substrate may affect the process of the transfer of proton/electron, thereby enhancing their re-activity of the substrate and/or reducing the barrier from a reaction”. Please see line 347-349 in the revised manuscript.

Question 11. English could be a lot better. I can follow the meaning of the sentences but only after reading them several times. The text requires extensive editing.

Answer: As suggested, we have extensively checked and revised the English language of the manuscript. Please see the revised manuscript for its clarity and readability.

Round 2

Reviewer 3 Report

The present manuscript (v2) is a much necessary improvement.  The text is now quite readable.  I do however have some problems with the Science: 

1. Is the position of 1-MP more stable in FIgure 1B than 1C?  If so do the authors think that the Phe residues play a role in the interconversion between the two bound states?

2. I still don't follow the message of Figure 3.  It appears that Phe 106 and 116 are not important for binding, while Phe207 is quite important even after Phe298 or Phe478 are mutated into Ala.  Phe298 and Phe478 are still quite important, but what effect can replacement with Ala298 have on Phe298 (green) since it is an Ala already?  equally what effect does Ala478 can have on Phe478 (blue) when it is already mutated into Ala?

3. I wonder if the reason why the MD and MM results in this manuscript cannot put the 1-MP into a reactive conformation of the CYP protein is because the temperature was not allowed to increase about 37 oC.  It is also possible that the available PDB structures that were used in this study are locked in an energy minimum that allows for crystallization but are not in an effective reactive conformation.  In solution studies such as NMR might be of used in understanding the problem.

Author Response

Questions from Reviewer #3:

Question 1. Is the position of 1-MP more stable in Figure 1B than 1C?  If so do the authors think that the Phe residues play a role in the interconversion between the two bound states?

Answer: The molecular docking is an in-silico method and only presents the potential conformations/interactions based on the conditions/parameters set in the study. Thus, the positions of 1-MP in Fig. 1B and 1C only reflect the probable conformations rather,  as only PHEs were set as flexible in the system. In the real world all residues can be flexible and the true conformations of 1-MP may be affected by other residues. Therefore, it is inappropriate to say that the position of 1-MP is more stable in Fig. 1B than 1C.

Moreover, we cannot fix a single residue in a certain protein in the real world under the currently available laboratory techniques. We thus applied in-silico molecular docking to help us understand the potential roles of some residues in the active site of an enzyme protein. As a result, Fig. 1 demonstrates the possible structures of 1-MP bound to CYP2E1 with different conditions of PHE, indicating the role of PHEs in the active site for the interaction between potential substrate and the enzyme.

Question 2. I still don't follow the message of Figure 3.  It appears that Phe 106 and 116 are not important for binding, while Phe207 is quite important even after Phe298 or Phe478 are mutated into Ala.  Phe298 and Phe478 are still quite important, but what effect can replacement with Ala298 have on Phe298 (green) since it is an Ala already?  equally what effect does Ala478 can have on Phe478 (blue) when it is already mutated into Ala?

Answer: In this study, we focused primarily on PHE298 and PHE478, as they have been reported to be “gatekeepers” of some tunnels through which some ligands may enter the active site. The F298A mutant did not show an obvious change in the energy contribution from the wild-type PHE298. But for F478A mutant, the ALA478 contributed a positive energy (a potential repulsive force), which suggests that the wild-type PHE478 may play an important role in ligand binding. Indeed, PHE207 in human CYP2E1 has been reported as an important amino acid residue for the binding of aromatic ligands. However, the energy contribution from PHE207 was not obviously changed following the mutation of  F298 or F478, indicating that the role of PHE207 in ligand binding might not be changed either. In addition, PHE207 is not a “gatekeeper” of tunnel 2b or 2c, both of which are the main tunnels for ligands to move in and out of the active site of CYP2E1.

Question 3. I wonder if the reason why the MD and MM results in this manuscript cannot put the 1-MP into a reactive conformation of the CYP protein is because the temperature was not allowed to increase about 37 oC.  It is also possible that the available PDB structures that were used in this study are locked in an energy minimum that allows for crystallization but are not in an effective reactive conformation.  In solution studies such as NMR might be of used in understanding the problem.

Answer: Thanks for reviewer’s constructive suggestion. However, the molecular docking and molecular dynamics were performed based on the provided parameters, including the properties of atoms, bonds, and non-bonding interactions, the structures of both ligand and protein, and other algorithms for solvent, coulombic or Lennard-Jones potentials and system environment. Thus, it is impossible for an increase in the temperature to promote a reaction, since the bonds between relevant atoms are constraints by the parameters. It can only accelerate the conformational change of ligand and protein for showing an allosteric effect. In addition, the details of a reaction between a ligand and an enzyme are usually studied via quantum chemistry approaches, which do not rely on the empirical parameters.

NMR has been widely used in the identification of chemical structures and can be used to study the protein smaller than 80kDa. However, NMR analysis is time- and labor-consuming, while our research team is mainly focused on metabolism-dependent toxicity of chemical using cultured cell lines or animal models. As a powerful technique, NMR is our close consideration in future studies.
